# A Torpedo Target Recognition Method Based on the Correlation between Echo Broadening and Apparent Angle

Zirui Wang [1],* , Jing Wu [2], Haitao Wang [2], Yukun Hao [1] and Huiyuan Wang [1]

1   School of Mechanical Engineering, Xi'an Jiaotong University, Xi'an 710049, China
2   Shanghai Electronic Ship Research Institute, China Shipbuilding Industry Corporation,
    Shanghai 201100, China
*   Correspondence: wzr.xajt@outlook.com

**Abstract:** As acoustic decoys can simulate the scale of the target through orderly control of the echo delay, simulated acoustic decoys have scale characteristics similar to those of the scaled target. Consequently, simulated acoustic decoys make it difficult for active acoustic homing torpedoes to recognize acoustic decoys through traditional echo broadening or apparent angle. This will lead to a decrease in the anti-interference capability of torpedoes. In combat, acoustic decoys deceive torpedoes and deviate from the tracking course so that torpedoes cannot find the real target, or waste the range, eventually failing to strike the target and failing in combat. The accurate underwater target scale recognition of active acoustic homing torpedoes is considered a difficult technique. In this paper, we propose a target recognition method based on the correlation between target echo broadening and apparent angle. This specific simulation example shows that conventional target scale recognition methods cannot distinguish between suspended and homing acoustic decoys with virtual scale. By contrast, the target scale recognition method proposed in this paper can accurately distinguish between suspended and homing acoustic decoys with virtual scale at close range, under non-positive transverse port angle conditions. This method improves the anti-interference capability of torpedoes. In addition, it can improve the accuracy of active sonar recognition scale targets of ships, which guide active sonar target recognition.

**Keywords:** target recognition; echo broadening; apparent angle; correlation

## 1. Introduction

The target echoes received by torpedoes are a collection of target reflection points, each of which independently reflects the incoming torpedo signal. The reflections from these points are not uniform. There are highly reflective points, usually called " highlights", such as a submarine's bow, hull, and stern [1,2]. These highlights have a relatively stable relative position and can characterize the target endpoint location; therefore, torpedoes usually obtain target scale information by extracting target echo highlight features. The scale of the target is stable and representative. As a stable feature, the scale of the target varies little with the environment, but the scale of different targets varies very significantly. Therefore, active acoustic homing torpedoes use the target highlights to estimate the scale of the target and use it as a criterion for target recognition, which has become the primary method for target recognition by active acoustic homing torpedoes (including various active sonars) [3–6]. Figure 1 shows the attack posture of the torpedo.

Many scholars have made great progress in the research of torpedo true and false target scale recognition. As the target scale is usually proportional to the target echo intensity, Lee D J et al. [7] converted acoustic measurements of echo intensity to fish scale estimates by constructing a database, while obtaining the relationship between target intensity and fish scale. Furthermore, a linear regression analysis was performed for all species to reduce the data to empirical equations, to show the variation of target intensity with fish scale and

---



species. On the other hand, Buerkle, U. [8] showed that the fish scale calculated from the echo intensity needed to be corrected when using the relationship between echo intensity and fish scale. This method did not consider the fish's orientation, since the active sonar receives the target echo intensity, which is not precisely proportional to the target scale, but to the reflected area of the target acoustic signal. In addition, the target echo strength received by the active sonar is also relative to the target material and structural ocean channel. Therefore, it is difficult for the active sonar to estimate the target scale using the target echo intensity for target recognition.

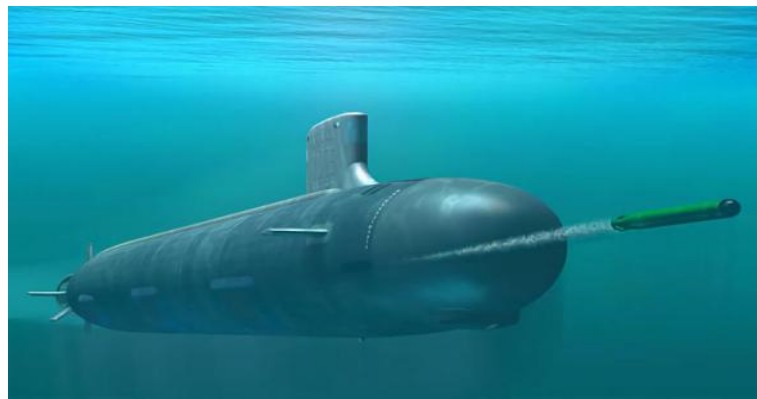

**Figure 1.** The torpedo attack posture.

Since it is challenging to recognize targets accurately based on the echo intensity, Wang Y et al. [9] recognized weak targets by target echo pulse broadening in a strong interference background. Moreover, they solved the problem of few echo features and the difficult recognition of weak targets underwater. They obtained the relationship between pulse broadening, target material, pulse power, and target distance of different underwater weak targets using statistical data analysis. The results show that the pulse-broadening characteristics of the echo signal can be used to recognize weak targets in the background of strong water impurity interference. Yu L et al. [10] estimated the echo broadening of the target, and derived the target length as a parameter by expectation maximization to obtain the feature parameters for recognizing the target. The simulation results show that the method has higher target recognition accuracy than other feature recognition methods. Xu Y et al. [11] recognized underwater targets by extracting underwater target orientation heading features. They used underwater target highlighting structure features to obtain underwater target orientation heading features using quadratic least-squares fitting. The experimental simulation results show that the target scale can be recognized by the target orientation heading features. However, when encountering strong interference, such as acoustic decoys with virtual scale, the active sonar makes a significant error in estimating the target scale. It is difficult to accurately identify the target by echo broadening and apparent angle (azimuthal orientation).

In this paper, we first simulate the echo signals received by torpedoes from submarines, suspended acoustic decoys, and mobile acoustic decoys for virtual-scale simulation. Secondly, we estimate the parameter of each target echo signal to obtain their echo broadening and apparent angles, which are converted into apparent scales, respectively. Then, we investigate the correlation between the echo broadening and the apparent angle of different targets. We thus propose a method to recognize the true and false targets based on the correlation between the target's echo broadening and apparent angle, which solves the problem that torpedoes are facing in recognizing the scale features of acoustic decoys simulating virtual scales. In addition, the method improves the ability of active sonar to perform target scale recognition with strong interference.

## 2. Basic Theory

### 2.1. Acoustic Decoy Scale Target Simulation

As it is difficult for a single point source acoustic decoy to cope with the scale recognition of torpedoes, acoustic decoys are currently used for scale simulation by controlling the echo response timing of each array element of multiple suspended acoustic decoys, or mobile acoustic decoy line array with communication capability, to achieve multi-highlight scale target simulation, so that the acoustic decoys have scale characteristics, or make their own scale characteristics increase [12,13].

Figures 2 and 3 are the schematic diagrams of suspended and mobile acoustic decoys simulating multiple highlights targets. After the acoustic decoys are estimated to obtain the torpedo distance, orientation, and other information, after mutual communication, each acoustic decoy (array element) carries out a certain delay, respectively, so that the torpedo detects the suspended acoustic decoys with a certain horizontal scale, and detects the mobile acoustic decoys with a horizontal scale larger than its own scale. The length of the detected virtual-scale target is determined by the simulated position of the acoustic decoys (array elements) perpendicular to the torpedo course, and the highlight distribution of the virtual-scale target is determined by the acoustic decoys (array elements) between the two ends of the acoustic decoys (array elements), where each acoustic decoy (array element) is simulated by their distance difference from the simulated position with time delay.

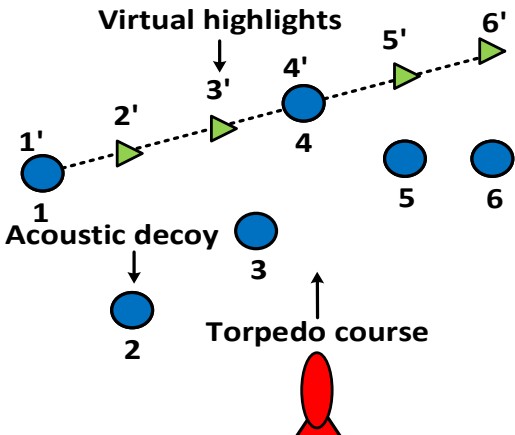

**Figure 2.** The suspended acoustic decoy simulating multi-highlight scale targets.

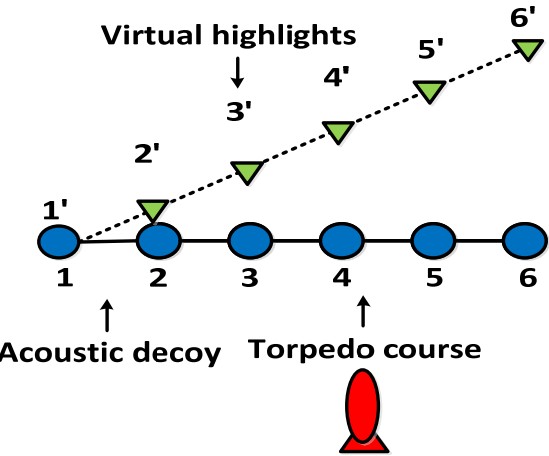

**Figure 3.** The mobile acoustic decoy simulating multi highlight scale targets.

### 2.2. Echo Broadening Estimation

The submarine echoes received by torpedoes usually have echo broadening characteristics due to the time delay generated by the submarine echoes along different parts of the radial direction [14,15]; the echo broadening of the submarine echo signal can be obtained from the echo highlight time delay by:

$$\tau = \tau_{max} - \tau_{min} - T \tag{1}$$

Among them, $\tau_{max}$ is the maximum value of highlight delay, $\tau_{min}$ is the minimum value of highlight delay, and $T$ is the pulse width of the torpedo launch signal.

The target echo signal will appear when the extension of the width, which is due to the sound waves at a certain angle on the reflector with a certain scale, such as submarines, in different parts, will cause the sound range difference:

$$\Delta\tau = \frac{2L\cos\varphi}{c} \tag{2}$$

Among them, the target length is $L$, and the angle between the incident wave and the target is $\varphi$, and the underwater sound speed is c, usually 1500 m/s. Figure 4 shows the geometric significance of the target echo broadening.

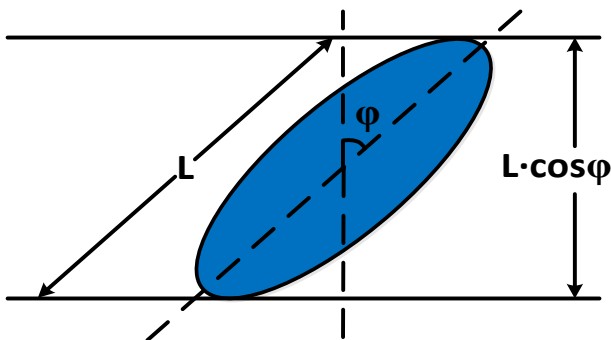

**Figure 4.** The target echo broadening.

### 2.3. Estimation of Apparent Angle

The submarine target has a certain degree of geometric size in space except for the specific angle of the gangway, and it is especially obvious in the lateral size. While large-scale targets have a certain variation pattern in orientation, the target spatial scale size can be obtained according to its obtained angle, the apparent angle; therefore, the apparent angle is an important parameter of torpedoes which is usually expressed by the difference between the first and last values of the azimuthal strike fitting slope [16,17]. Figure 5 shows the split echo signal.

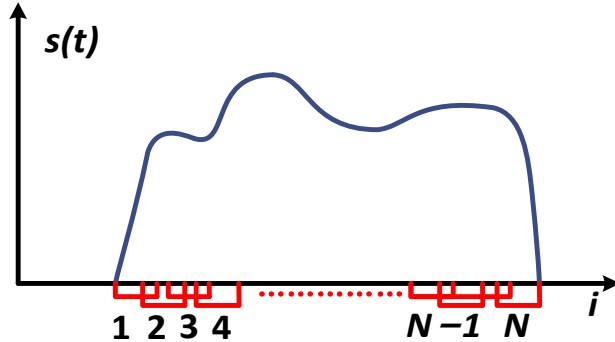

**Figure 5.** The split echo signal.

The apparent angle is extracted by splitting the received echo signal into $N$ subframes in the time domain, which is shown in the figure above. The exact orientation of the split beam is obtained for each subframe signal $\theta_i$ ($i$ = 1, 2, $\cdots$, $N$). The least-squares fitting [18,19] is performed for each of the obtained orientations, and the fitted linear equations are as follows:

$$\theta = a + b \cdot i \tag{3}$$

Among them, $i$ is the number of subframes and $\theta$ is the corresponding azimuth value of each subframe. The least-squares method needs to minimize the fitting variance $\Theta$:

$$\Theta = \sum_{i=1}^{N} (\theta_i - \theta)^2 = \sum_{i=1}^{N} (\theta_i - (a + b \cdot i))^2 \tag{4}$$

The following conditions need to be met:

$$\frac{\partial \Theta}{\partial a} = 0 \Rightarrow \sum_{i=1}^{N} (-2\theta_i + 2a + 2bi) = 0$$
$$\frac{\partial \Theta}{\partial b} = 0 \Rightarrow \sum_{i=1}^{N} (-2\theta_i i + 2ai + 2bi^2) = 0 \tag{5}$$

Set $\sum_{i=1}^{N} \theta_i = A$, $\sum_{i=1}^{N} \theta_i \cdot i = B$. Then, the intercept and slope of the fitted line can be derived:

$$a = \frac{6}{N(N+1)} \left( \frac{2N+1}{3} A - B \right)$$
$$b = \frac{12}{N(N^2-1)} \left( B - A \frac{N+1}{2} \right) \tag{6}$$

The apparent angle $T_r$ is obtained:

$$T_r = \theta(N) - \theta(1) = b(N - 1)$$
$$= 12 \left( \frac{B}{N(N+1)} - A \frac{1}{2N} \right) \tag{7}$$

Figure 6 shows the schematic diagram of the orientation approach method.

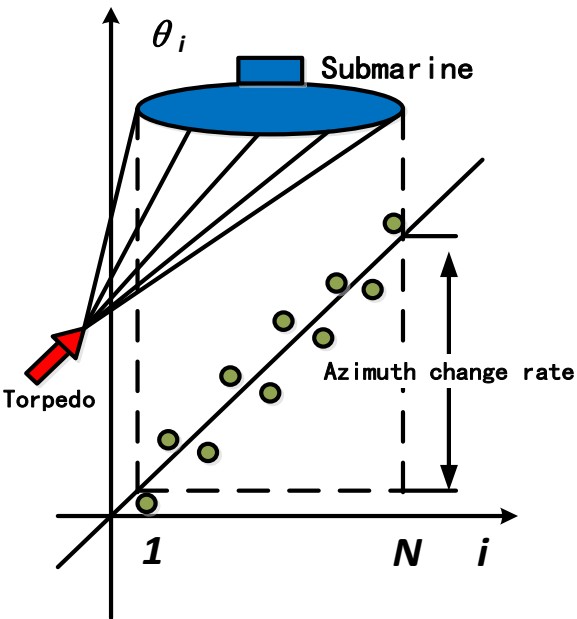

**Figure 6.** The azimuth direction method.

### 3. Correlation Analysis of Echo Broadening and Apparent Angle

The main scale feature parameters currently used by torpedoes to recognize targets are the two mentioned above; echo broadening and apparent angle. The acoustic decoys can be simulated, making it difficult to recognize the torpedoes. Since the echo broadening and apparent angle characterize the target scale characteristics in the time and angle domain, they are both characterizing the same target characteristics; therefore, there must be a logical correlation between them. In this paper, we propose a scale recognition algorithm, based on the correlation between target echo broadening and apparent angle, which converts target echo broadening and apparent angle into apparent scale, respectively, and recognizes the target according to the difference between the two apparent scales. This scale recognition method further enhances the recognition of a logical relationship between scale features, based on the traditional single-feature parameter recognition method. Hence, the torpedoes may still recognize the acoustic decoys due to their scale features' inability to match, even though they simulate the submarine echo broadening and apparent angle on a large scale. This method improves the ability of the torpedo scale to recognize targets.

The apparent tensor is the azimuthal range of the echoes received by the torpedo from different parts of the target combined with the target distance; the scale at which the torpedo detects the target, which is the apparent scale, can be obtained as follows:

$$L = 2R \cdot \sin \frac{\alpha}{2} \tag{8}$$

Among them, $L$ is the apparent target scale, $\alpha$ is the apparent angle, and $R$ is the target distance. Figure 7 shows the geometric relationship between apparent scale and apparent tensor correlation.

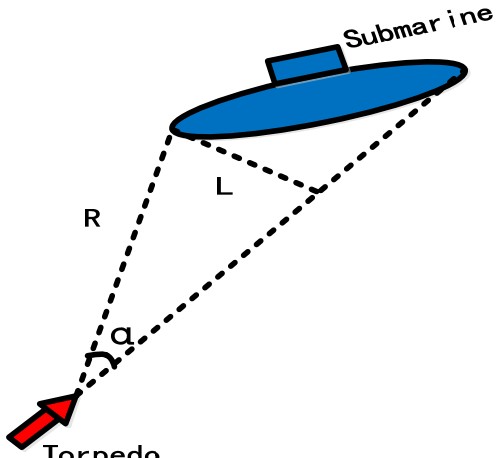

**Figure 7.** The association between apparent scale and apparent angle.

The echo broadening is the time delay difference between the echoes received by the torpedo from different parts of the target, combined with the target attitude angle. The apparent target scale can be obtained as follows:

$$L = \frac{D \cdot \tan \theta \cdot c}{2} \tag{9}$$

Among them, $D$ is the target echo broadening, $\theta$ is the target side angle obtained by estimating the position of the target in the previous cycle and the current cycle, and the underwater sound speed is c, usually 1500 m/s. Figure 8 shows the geometric relationship between the apparent scale and the echo broadening correlation.

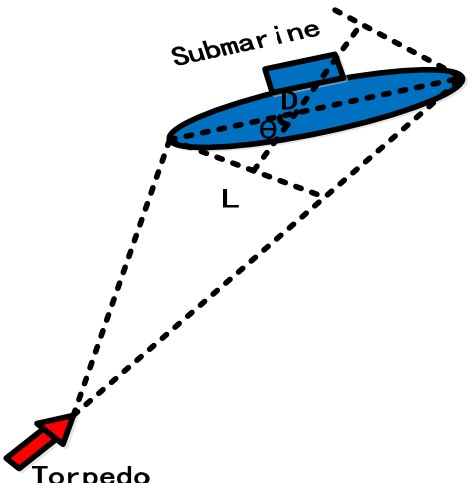

**Figure 8.** The correlation between apparent scale and echo broadening.

### 4. Simulation Verification

The echo of any target is composed of several sub-echoes iteratively; each sub-echo can be regarded as emitted from a scattering point, which is the bright spot. Therefore, we can equate any target to a combination of several bright spots. The acoustic decoys are usually modeled as multi-highlight echoes to simulate scale targets [20,21].

When the echo simulation is performed by suspended acoustic decoys, the virtual-scale target simulation is achieved by simulating the echoes of the bright spots of the suspended acoustic decoys at both ends of the torpedo course plumb line; in addition, the time delay simulation amount of the rest of the acoustic decoys is determined according to their distance from the bright spots of their corresponding virtual-scale targets. When the echo simulation is performed by the mobile acoustic decoy line array, each array element performs delay superposition and echo time broadening on the torpedo-seeking signal, according to the incoming torpedo posture, respectively, to realize the virtual-scale simulation of the submarine target.

Among them, the bright spot echo model of a single suspended acoustic decoy and a single array element of a homing acoustic decoy array can be determined based on the three parameters of amplitude factor, time delay, and phase jump. The transfer function can be expressed as:

$$H_i(r, \theta, \psi, \omega) = A_i(r, \theta, \psi)e^{j\omega(v)\tau_i}e^{j\varphi} \tag{10}$$

where $r$ is the distance from the torpedo to the target bright spot; $A_i(r,\theta,\Psi)$ is the amplitude of the echo of this bright spot, which is related to the distance $r$ of the target and the direction of acoustic wave incidence, i.e., the incidence angle $\theta$ and the pitch angle $\Psi$. $\tau_i$ denotes the time delay of this bright spot, determined by the acoustic range of the equivalent acoustic center concerning some reference point, and is a function of $\theta$. $\omega(v)$ denotes the difference between the echo's center frequency and the incident wave's center frequency, due to the target's relative motion by a Doppler frequency shift; $\varphi$ is the phase jump of the return wave.

Therefore, the total transfer function of the target is:

$$H_i(r, \theta, \psi, \omega) = \sum_{i=1}^{N} A_i(r, \theta, \psi)e^{j\omega(v)\tau_i}e^{j\varphi} \tag{11}$$

where $N$ is the number of target highlights.

As the plate element analysis method is capable of accurate prediction of underwater complex target geometry backscattering, we built a benchmark submarine scattering model by plate element method [22,23]. Based on the high-frequency approximation of Kirchhoff's

integral formula, we can divide the target surface into plate elements, calculate the scattered sound field for each plate element separately, and obtain the whole scattered sound field after superposition.

The Kirchhoff formula for the acoustic scattering problem is [24]:

$$\varphi_s(r_2) = \frac{1}{4\pi} \iint_S [\varphi_s \frac{\partial}{\partial n} \left( \frac{e^{jkr_2}}{r_2} \right) - \frac{\partial \varphi_s}{\partial n} \frac{e^{jkr_2}}{r_2} ] ds \tag{12}$$

where $s$ is the scatterer surface, $n$ is the surface outer normal, $r_2$ is the scattering point vector diameter, $\varphi_s$ is the scattering potential function, and $k$ is the wave vector.

The Kirchhoff approximation is a high-frequency approximation that typically assumes that:

The contribution of the geometric shadow region to the scattered field is neglected.

The object surface satisfies the rigid boundary conditions. Specifically, the scattering potential function under rigid boundary conditions is equal in size to the incident potential function, and the angle is symmetric about the target:

$$\begin{cases} |\varphi_s| = |\varphi_i| \\ \frac{\partial (\varphi_s + \varphi_i)}{\partial n} = 0 \end{cases} \tag{13}$$

where the incident wave potential function $\varphi_i = (A/r_1)\exp(ikr_1)$ (omitting the time factor $e^{-j\omega t}$), and A is the amplitude constant. In the case of transceiver co-location, the surface boundary conditions lead to:

$$\varphi_s = \frac{-ik_0 A}{2\pi} \int_{S_0} \frac{e^{i2k_0 r}}{r^2} \cos \theta ds \tag{14}$$

where $r$ is the scattering point vector diameter and $\theta$ is the angle of incidence. We extend the above results to non-rigid surfaces. When the surface radius of curvature $R$ is large (the product of the radius of curvature and wavelength is much larger than 1), the local plane wave approximation can be applied by setting the surface reflection coefficient to $V(\theta)$, and the surface acoustic impedance to $Z_n$:

$$\begin{cases} |\varphi_s| = |V(\theta)| \cdot |\varphi_i| \\ \frac{i\omega \rho (\varphi_s + \varphi_i)}{\partial (\varphi_s + \varphi_i)/\partial n} = -Z_n \end{cases} \tag{15}$$

where $\omega$ is the incident wave angular frequency and $\rho$ is the target surface density. The following equation relates the plane wave reflection coefficient to the total surface acoustic impedance of any complex interface at an infinitely large plane interface:

$$\frac{\rho c / \cos \theta}{Z_n} = \frac{1 - V(\theta)}{1 + V(\theta)} \tag{16}$$

where the underwater sound speed is c, usually 1500 m/s. Equation (15) can be substituted for Equation (16). In the case of transceiver co-location, the surface boundary conditions lead to:

$$\varphi_s = \frac{-ik_0 A}{2\pi} \int_{S_0} \frac{e^{i2k_0 r}}{r^2} V(\theta) \cos \theta ds \tag{17}$$

We can obtain the target intensity in the far-field condition according to Equation (17):

$$TS = 10 \log \left| -\frac{ik_0}{2\pi} I \right| \tag{18}$$

Among them:

$$I = \int_{S_0} e^{2ik_0 \vec{\rho} \cdot \vec{r_0}} (\vec{n_0} \cdot \vec{r_0}) V(\theta) \mathrm{d}s \tag{19}$$

Among them, $\vec{\rho} = x\vec{i} + y\vec{j}$ is the vector from the point where the face element is located to the reference point, $\vec{n_0} = \vec{k}$ is the unit normal vector of the face element, $\vec{r_0} = u\vec{i} + v\vec{j} + w\vec{k}$ is the unit vector from the receiving point to the reference point, and $\vec{n_0} \cdot \vec{r_0} = \cos\theta = w$. Suppose that the reflection coefficient $V(\theta)$ is constant within a plate element; then, the integral of a plate element is:

$$I_{\Delta s} = \int_{\Delta s} e^{2ik_0(ux+vy)} V(\theta) \mathrm{d}x\mathrm{d}y = V(\theta)w \sum_{n=1}^{K} \frac{e^{2ik_0(x_n u + y_n v)}(p_{n-1} - p_n)}{(2k_0 u + 2k_0 p_{n-1} v)(2k_0 u + 2k_0 p_n v)} \tag{20}$$

where $K$ is the number of polygon vertices and $(x_n, y_n)$ is the coordinate of the polygon vertex and set $p_0 = \frac{y_1 - y_K}{x_1 - x_K}$.

The target intensity of a complex target can be calculated by the following slab metric method. The target surface is first divided into a grid of many small slabs. The scattered sound field of all the plate elements in the bright area is summed to obtain the approximate value of the scattered sound field of the target, and the plate elements with different orientations are transformed uniformly to some determined plane by coordinate transformation; consequently, we can ascertain:

$$I = \sum_{i=1}^{N} \sum_{j=1}^{M} \left[ V(\theta_{ij}') w' \sum_{n=1}^{K} \frac{e^{2ik_0(x_n' u_{ij}' + y_n' v_{ij}')}(p_{n-1} - p_n)}{(u_{ij}' + p_{n-1} v_{ij}')(u_{ij}' + p_n v_{ij}')} \right] \Big|_{S_{ij}} \tag{21}$$

where $x_n', y_n'$ is the coordinate of the vertex of the $(i,j)$th plate after transformation to the 2D plane, $V(\theta_{ij}')$ is the local surface reflection coefficient of the $(i,j)$th plate, and $\theta_{ij}'$ is the angle between the normal and incident sound lines of the $(i,j)$th plate. In the later specific calculations, the triangular plate element is adopted; therefore, $K = 3$ after a simulated Doppler frequency shift.

Figure 9 shows the benchmark submarine target echo. We can see the relationship between the target intensity of each part of the submarine (transom, enclosure, and bow), obtained by matching the filtering gain with the angle. The target intensity of the submarine is more significant when the angle is 0°–90°, because the frontal scattering area of the submarine is large, and the structure is simple; the phase cancellation is less. If the angle is closer to 90°, the time delay difference of the target echoes in each part of the submarine are smaller.

Figure 10 shows the torpedo and target movement posture. As the fixed advance angle guidance method is less difficult in engineering practice, and can make the torpedo attack end effective, it is an ideal guidance method for an active acoustic homing torpedo. The attack posture of the torpedo can be set using the fixed advance angle guidance method; $O$ point is the active acoustic homing torpedo initial position, $A_0$ point is the target initial position, the distance $r_0$ between the torpedo and the target is 500 m, the orientation $\theta$ of the target initially located in the torpedo is 45°, the torpedo speed $v_t$ is 50 kn, and the torpedo fixed advance angle $\alpha$ is 5°. Since the torpedo usually attacks from a certain angle behind the side of the target, we can set the initial enemy side angle $\Phi$ to 135°. When the target is a submarine or a homing acoustic decoy, the target velocity $v_m$ is 15 kn, the submarine is 60 m, and the homing acoustic decoy line array is 40 m. Ten suspended acoustic decoys are deployed at point $O$; when the targets are suspended acoustic decoys, the deployment scatter error is 40 m [25].

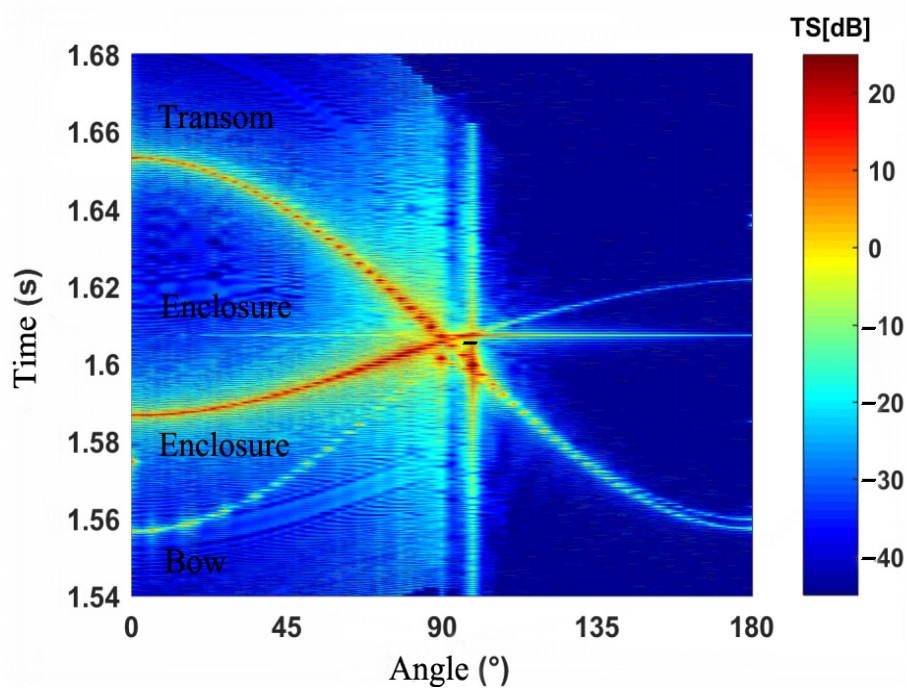

**Figure 9.** The benchmark submarine target echo.

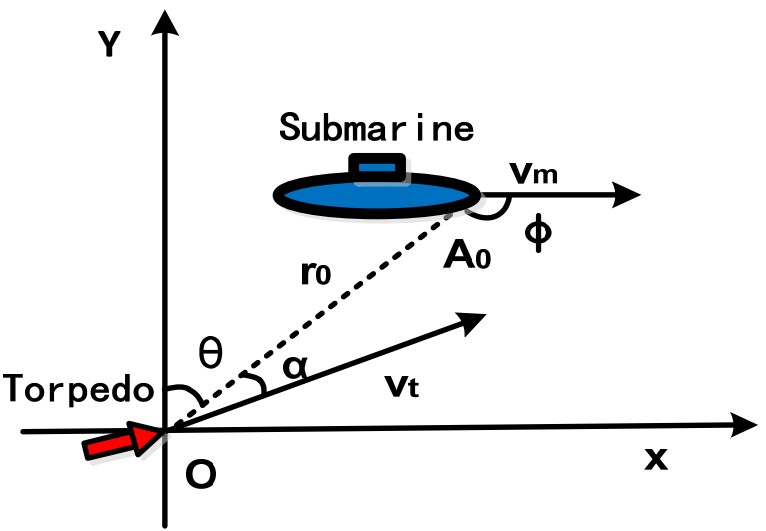

**Figure 10.** The torpedo and the target movement posture.

The operating frequency of the torpedo transmitting signal was set to be 30 kHz and bandwidth to be 0.8 kHz. The signal period was 1 s, the time width was 62.5 ms, the sound source level 220 dB, the noise level 60 dB, the receiving directivity index 17 dB, and the detection index 12 dB.

When the acoustic signal propagates in the ocean channel, the surface and bottom reflections produce multipath scattering [26–29]. Moreover, the sound velocity gradient causes the refraction of acoustic waves, which tend to refract toward the lower sound velocity region, leading to the propagation of the acoustic signal curve. This paper applies Dushaw's depth-dependent sound velocity profiles derived from summertime temperature, pressure, and salinity data in the North Atlantic off the coasts of Britain and Ireland [30]. Figure 11 shows the sound speed profile data. The refraction and reflection of the acoustic signal cause the acoustic signal to propagate in different channels, causing the attenuation of the single-channel acoustic signal amplitude and the superposition of the different

channel acoustic signal time delays. The target echo simulation was performed, using the BELLHOP ray tracing program, to obtain the time delay and amplitude attenuation of the target echo signals of different channels. We superimpose them to obtain the multipath echo signals. The target scattering was simplified to specular scattering in the simulation because the target scattering angle was difficult to simulate. Figure 12 shows the acoustic signal propagation trajectory obtained by simulation using the BELLHOP program. The depth of the ocean is set at 600 m. The depth of the sound source and the receiving point is 200 m, and the distance between them is 500 m. The angle range of the sound source signal emission is $-15°$–$15°$, and the sound source emits 20 signals at equal intervals. The receiving point receives signals in the depth range of its depth $-10$ m–10 m, and the receiving angle range is also $-15°$–$15°$. Our simulation of the acoustic signal propagation path can make the echo signal more realistic, which in turn makes the verification of the effect of the proposed target recognition method more reliable.

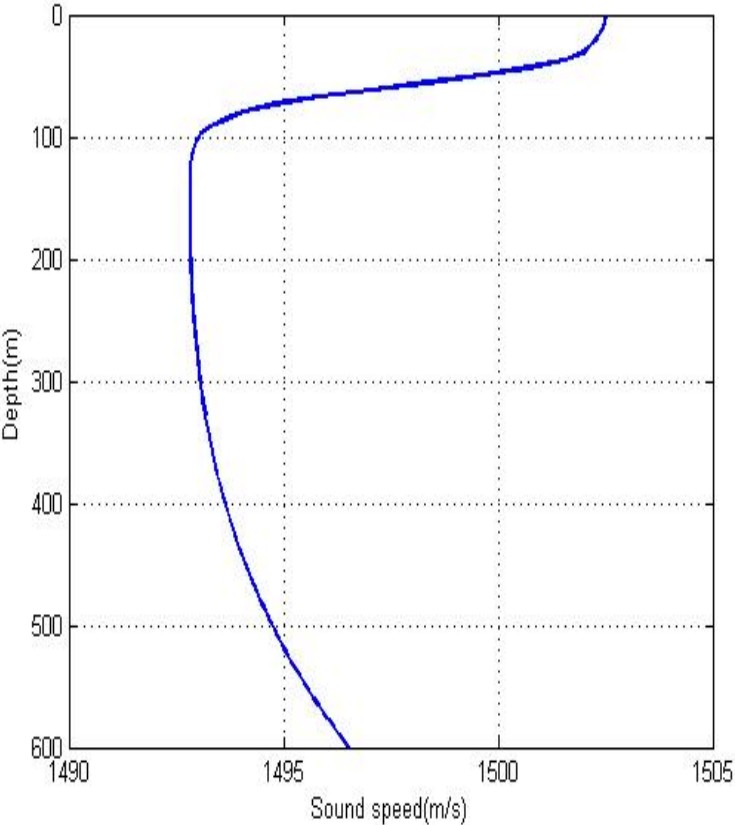

**Figure 11.** The sound speed profile.

Among them, Figure 13a–d are the estimation process of the target distance, the orientation, the apparent angle, and the echo broadening information, respectively [31]. Then, the apparent angle and echo broadening of each cycle are converted into the apparent scale to analyze the correlation between echo broadening and apparent angle of the submarine, mobile acoustic decoy, and suspended type acoustic decoy.

Figure 14 shows the apparent scale of the target converted according to the apparent angle and echo broadening of each cycle. The echo broadening and apparent angle cannot distinguish the scale difference between the real-scale target and the virtual-scale target. Under substantial interference, the target scale cannot be accurately recognized only by the echo broadening and apparent angle alone. Therefore, we recognized the target by correlating the echo broadening and the apparent angle. We also analyzed the difference between the equivalent scales obtained from the target echo broadening and the apparent angle in the following.

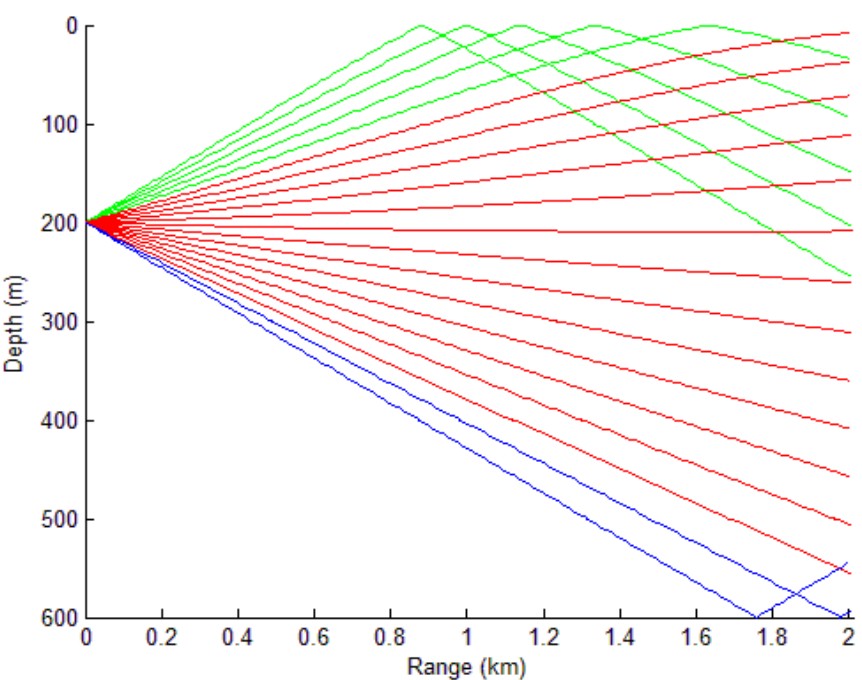

**Figure 12.** The underwater acoustic signal propagation.

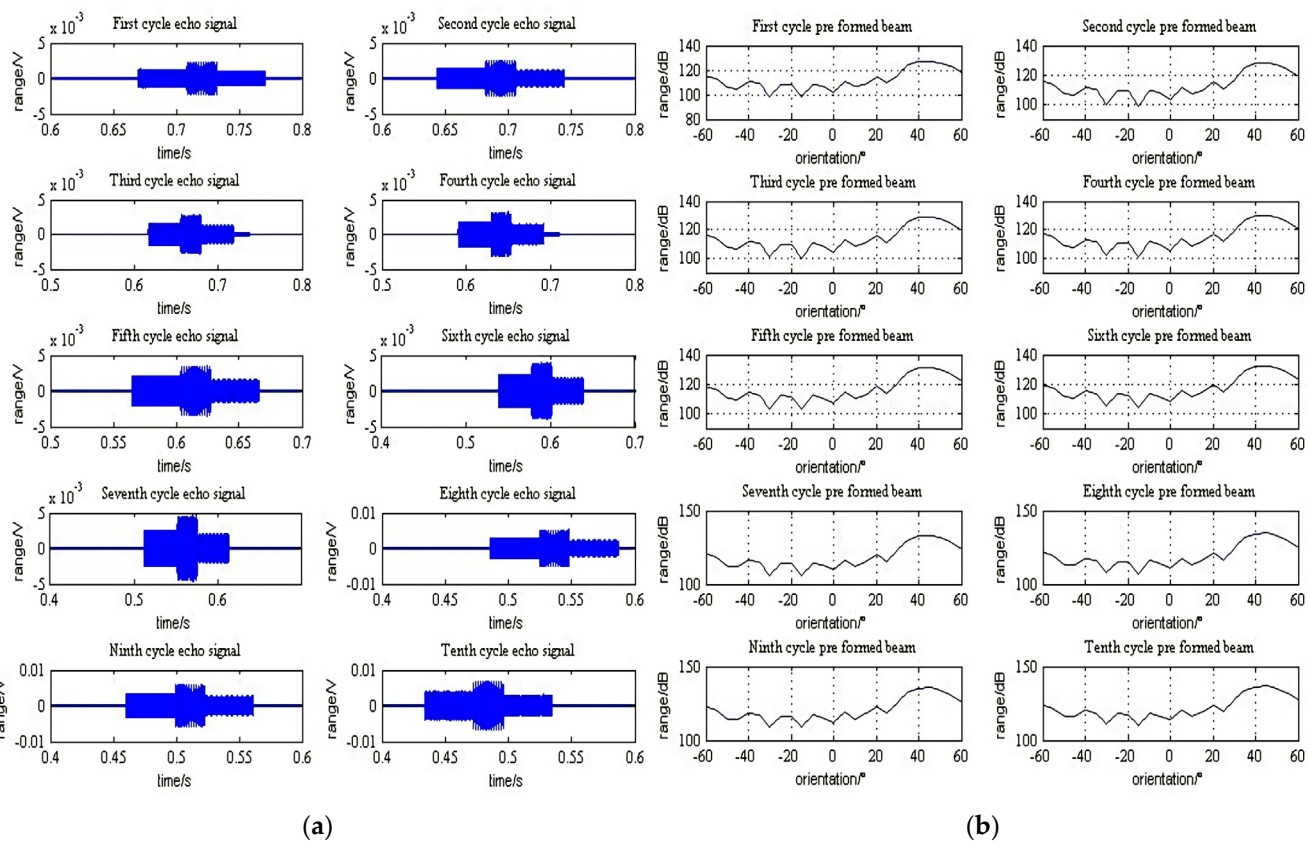

(**a**)                                                 (**b**)

**Figure 13.** *Cont.*

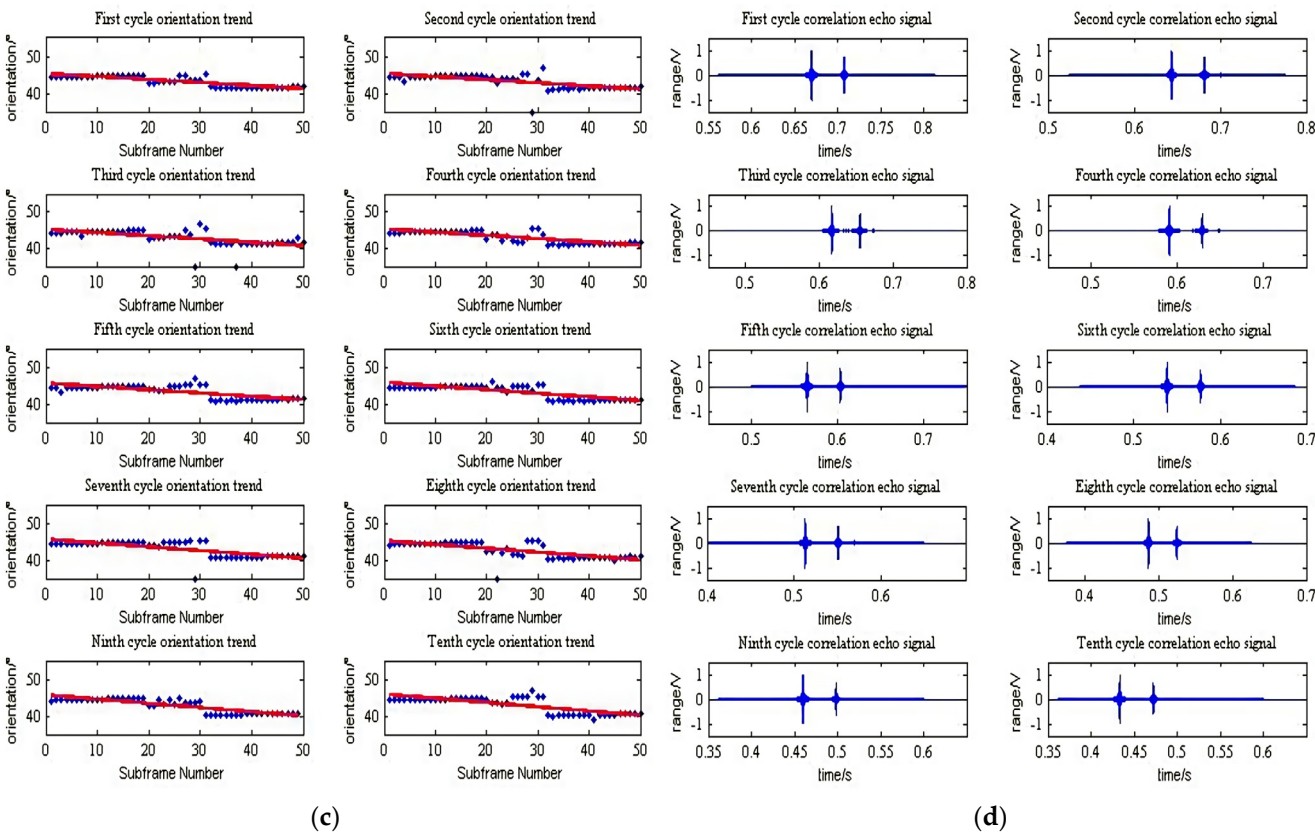

(**c**)  (**d**)

**Figure 13.** The target echo signal parameter estimation process. (**a**) The time domain echo signal. (**b**) The azimuth pre-beamforming. (**c**) The target orientation. (**d**) The time domain signal after correlation.

The apparent scale difference of each target is obtained from the apparent angle and the echo broadening; as shown in Figure 15, it can be observed that the apparent scale difference of each cycle of the submarine target is within the range of [0.2 m, 4.0 m], which is obviously smaller than the apparent scale difference of the mobile acoustic decoy and the suspended acoustic decoy.

Figure 16 shows the apparent scale difference of different ship angles. The target porthole angle is set to be 0°–180° in steps of 15°. It can be seen that when the target angle is in the range of 0°–75° and 105°–180°, the apparent scale difference of the submarine target is in the range of [0.5 m, 2.2 m], which is much smaller than the apparent scale difference of the mobile acoustic decoys and suspended acoustic decoys. When the target angle is in the range of 75°–105°, the apparent scale difference of submarine, mobile acoustic decoys and suspended acoustic decoys is about 12 m, so it is impossible to distinguish between the targets. Therefore, the torpedoes in this chord angle range cannot recognize the real and fake targets based on the echo broadening and apparent angle. This is due to the fact that the time delay difference between the echoes at the two ends of the target is very small near the positive transverse chord angle, which corresponds to the small equivalent distance between them. In addition, the echoes have mostly overlapped, and it is difficult to distinguish the bright spots by subframe segmentation, thus causing errors in accurate orientation estimation and making the spatial scale of the simulation smaller than the theoretical spatial scale at this time, resulting in a large error.

Figure 17 shows the apparent scale difference at different initial distances. The initial target distance was set from 500 m–2000 m, with 500 m as a step. It can be seen that when the target distance is 500 m, the apparent scale difference between the submarine target is 2.4 m, which is much smaller than the apparent scale difference between the mobile and suspended acoustic decoys. When the target distance is greater than 1000 m, the

apparent scale difference between the submarine, mobile and suspended acoustic decoys is very small, and it is impossible to distinguish between the targets; therefore, the torpedo cannot recognize the real and fake targets according to the echo broadening and apparent angle in the chord angle range. When the target is outside the scale recognition range, the echo signal propagation loss is larger and the signal-to-noise ratio of the target echo signal is lower, which leads to the inability of the torpedo to estimate the target parameters accurately. Moreover, the multipath effect exists in the long-range target echo, resulting in an inevitable time overlap of the target echo signals from different channels. As the estimated target echo broadening of the torpedo is larger than the time extension, due to the scale of the target itself, the target echo broadening cannot be estimated accurately, and it cannot recognize the target by the target scale recognition method proposed in this paper.

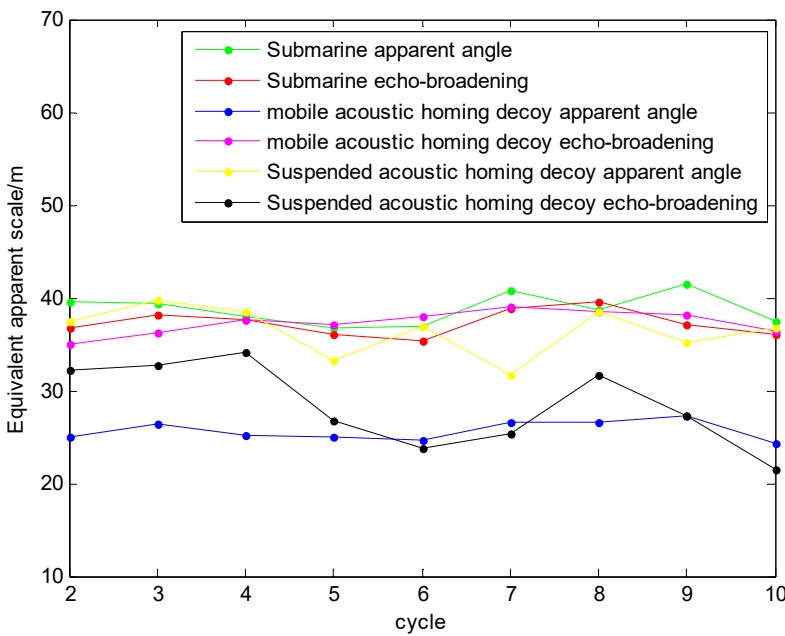

**Figure 14.** The equivalent apparent scale of each cycle.

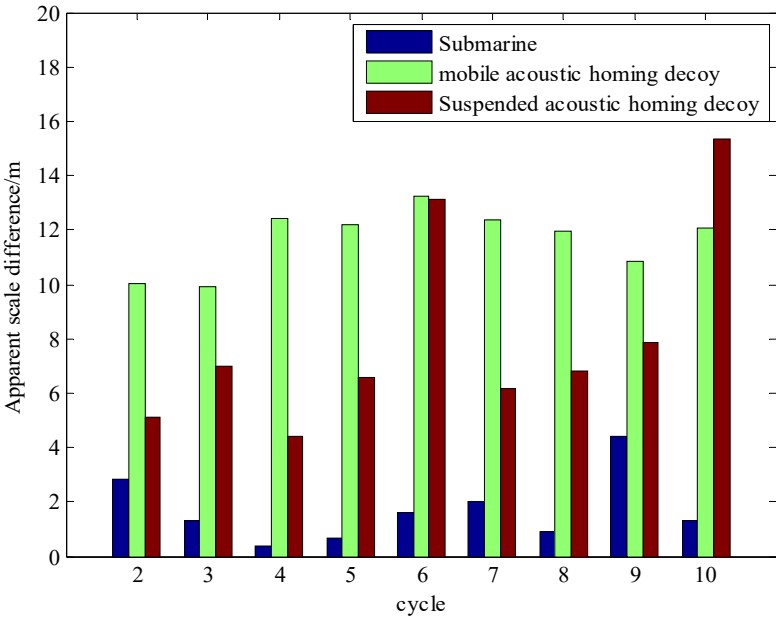

**Figure 15.** The apparent scale difference of each period.

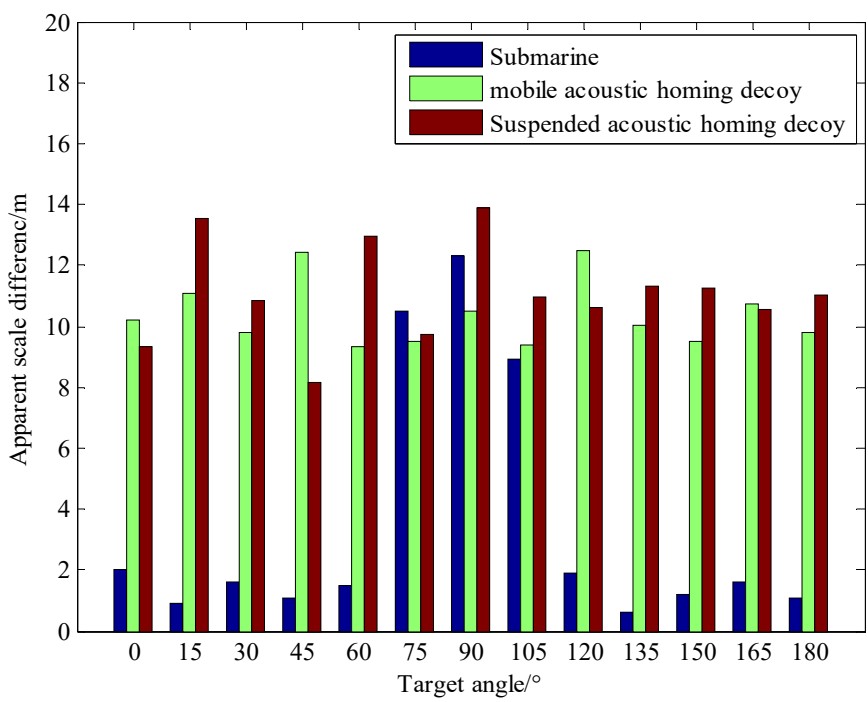

**Figure 16.** The apparent scale difference of different ship angles.

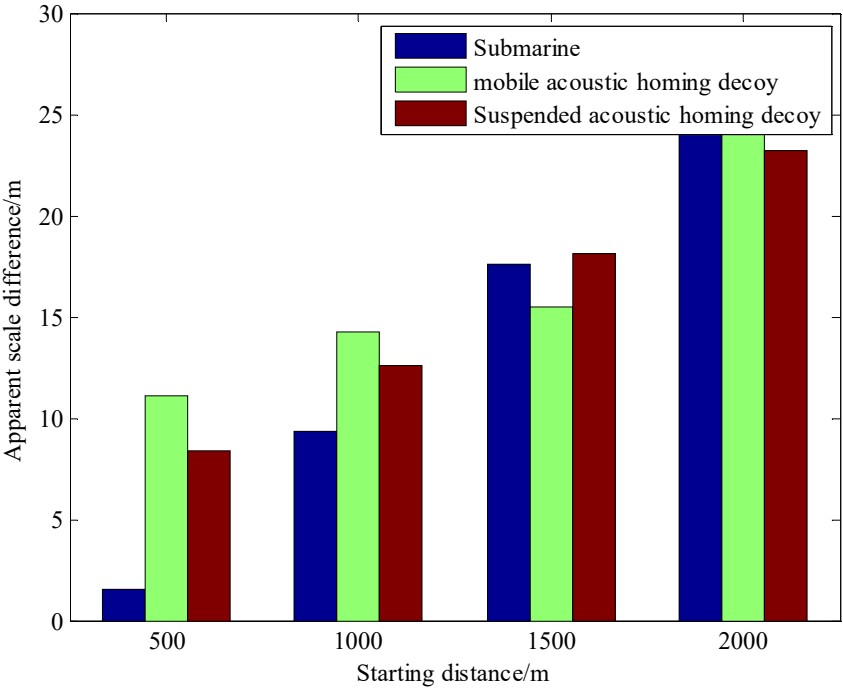

**Figure 17.** The apparent scale difference at different initial distances.

## 5. Conclusions

This paper addresses the problem that it is difficult for torpedoes to recognize mobile and suspended acoustic decoys by simulating virtual scales based on the single feature parameters of echo broadening and apparent angle. We propose a target scale recognition method, based on the correlation between echo broadening and apparent angle, which improves the ability of torpedoes to recognize targets and thus improves the accuracy of

torpedo attack targets. Due to the problems of signal-to-noise ratio, multipath effect, and superposition of echo signals, the torpedo estimated target echo broadening and apparent angle accuracy (at a long distance (about 500 m away)) is poor. Thus, applying the method produced by the correlation of echo broadening and apparent angle to recognize the target is not easy at low signal-to-noise ratios. As the time delay difference between the two ends of the target echoes is very small near the sinusoidal angle, the equivalent distance between them is very small. The echoes mostly overlap, and it is difficult to distinguish the bright spots by subframe segmentation, resulting in poor accuracy in estimating the apparent angle of the torpedo in the range of 75°–105°. Therefore, when the target distance and the outboard angle are within 500 m and 0°–75°,105°–180°, respectively, the torpedo can recognize the true and false targets using the echo broadening and apparent angle correlation. The distance range at which the torpedo can carry out accurate target scale recognition is related to the ocean environment, sea surface and seafloor undulation, ocean depth, target navigation depth and other factors, and can be farther under deep sea conditions. The target recognition method proposed in this paper has stronger anti-interference capability than the traditional target recognition method based on a single echo broadening or apparent angle, and applies to all kinds of active underwater sonar.

The target scale recognition method, based on the correlation between echo spreading and apparent angle, can also improve the accuracy of target scale recognition of ships. Our next step is to apply this torpedo-accurate target recognition method to general ships for recognizing other ships and reefs, and to fishing ships for recognizing fish size.

**Author Contributions:** Writing—original draft preparation, Z.W.; writing—review and editing, Z.W., J.W., H.W. (Haitao Wang), H.W. (Huiyuan Wang) and Y.H.; conceptualization, Z.W. and J.W.; methodology, Z.W. and J.W.; funding acquisition, H.W. (Haitao Wang); software, J.W.; validation, Z.W., H.W. (Haitao Wang), H.W. (Huiyuan Wang) and Y.H.; supervision, H.W. (Haitao Wang) All authors have read and agreed to the published version of the manuscript.

**Funding:** This research was funded by the underwater multi-physics field high-precision intelligent simulation and target recognition new system research and development program, grant number C3394BEF.

**Institutional Review Board Statement:** Not applicable.

**Informed Consent Statement:** Not applicable.

**Data Availability Statement:** Not applicable.

**Conflicts of Interest:** The authors declare no conflict of interest.

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
