# Peer review of "A Torpedo Target Recognition Method Based on the Correlation between Echo Broadening and Apparent Angle"

_applsci, doi:10.3390/app122312345_

Round 1

Reviewer 1 Report

This paper has proposed a torpedo target recognition method based on the correlation between target echo-broadening and apparent angle.

I think this paper can be accepted at the current version. However author can cite the following paper

1. Lee, D. H., Shin, J. W., Do, D. W., Choi, S. M., & Kim, H. N. (2017). Robust LFM target detection in wideband sonar systems. IEEE Transactions on Aerospace and Electronic Systems, 53(5), 2399-2412.

2. Komari Alaie, H., & Farsi, H. (2018). Passive sonar target detection using statistical classifier and adaptive threshold. Applied Sciences, 8(1), 61.

3. Samui, P., Roy, S. S., & Balas, V. E. (Eds.). (2017). Handbook of neural computation. Academic Press.

4. Boyell, R. L. (1976). Defending a moving target against missile or torpedo attack. IEEE Transactions on Aerospace and Electronic Systems, (4), 522-526.

5. Lee, K. C., Roy, S. S., Samui, P., & Kumar, V. (Eds.). (2020). Data Analytics in Biomedical Engineering and Healthcare. Academic Press.

Author Response

Dear Prof:

Thank you for your very insightful comment. This letter records our responses to your comments on our manuscript entitled "A torpedo target recognition method based on the correlation between echo-broadening and apparent angle" (ID: applsci-2038914).

Point 1: This paper has proposed a torpedo target recognition method based on the correlation between target echo-broadening and apparent angle.

I think this paper can be accepted at the current version. However author can cite the following paper:

  1. Lee, D. H., Shin, J. W., Do, D. W., Choi, S. M., & Kim, H. N. (2017). Robust LFM target detection in wideband sonar systems. IEEE Transactions on Aerospace and Electronic Systems, 53(5), 2399-2412.
  2. Komari Alaie, H., & Farsi, H. (2018). Passive sonar target detection using statistical classifier and adaptive threshold. Applied Sciences, 8(1), 61.
  3. Samui, P., Roy, S. S., & Balas, V. E. (Eds.). (2017). Handbook of neural computation. Academic Press.
  4. Boyell, R. L. (1976). Defending a moving target against missile or torpedo attack. IEEE Transactions on Aerospace and Electronic Systems, (4), 522-526.
  5. Lee, K. C., Roy, S. S., Samui, P., & Kumar, V. (Eds.). (2020). Data Analytics in Biomedical Engineering and Healthcare. Academic Press.

Response 1: Thank you for your encouraging comment! Yes, citing these papers gives our research content a more solid theoretical foundation and is more convincing. We accept your comment and have added them at the References section according to the article's content. And we refer to it when we perform a linear fit to the orientation direction in line 142 and when we simulate the torpedo-target countermeasure posture in line 280. Moreover, we refer to it when simulating the matched filtering gain results in line 318.

Thank you for reading our responses and reviewing the revised manuscript!

Best regards.

Yours sincerely,

Zirui Wang.

Name: Zirui Wang.

E-mail: [email protected].

Reviewer 2 Report

The authors have proposed the interesting method for recognizing the echoes produced by the real and false (decoy made) echoes within the the echo-location issues. However some improvements must be implemented to level the paper quality.

1- The authors should apply ':' indstead'.' when they begin to describe the sybbols within the formulas.

2- Within the formulas in lines 93-95 and 151-152 the velocity parameter c is not explained.

3- Lines 93 - 95 refer to Figure 4. and refer to the angle fi while at the Figure 4 the angle theta is shown.

4- In Line 128 the sentence'target scale at the time and air domain' should probably sound: '.. in the time and angle domain'.

5- Within the lines 170-175 the authors cite the foreign results of the signal analysis included in papers [18] & [19]. The description of presented Figure 9. is too short and moreover the cited figure has no labels at the axes.

6- The description of the configuration target-torpedo stated in lines 185 - 186 is unclear and needs to be improved.

Author Response

Dear Prof:

Thank you for your very insightful comment. This letter records our responses to your comments on our manuscript entitled "A torpedo target recognition method based on the correlation between echo-broadening and apparent angle" (ID: applsci-2038914).

Point 1: The authors have proposed the interesting method for recognizing the echoes produced by the real and false (decoy made) echoes within the the echo-location issues. However some improvements must be implemented to level the paper quality.

The authors should apply ':' indstead'.' when they begin to describe the sybbols within the formulas.

Response 1: Thank you for the comment. We have replaced the '.' with a ':' when we begin to describe the sybbols within the formulas.

Point 2: Within the formulas in lines 93-95 and 151-152 the velocity parameter c is not explained.

Response 2: Thank you for the comment. We have added the explanation of the parameter c. The modified parameters c in the article indicate the underwater sound speed in lines 124-125 and 185.

Point 3: Lines 93 - 95 refer to Figure 4. and refer to the angle fi while at the Figure 4 the angle theta is shown.

Response 3: Thank you for the comment. We have modified θ in Figure 4 in lines 126-127 to φ so that it is the same as the interpretation of the image in line 125.

Point 4: In Line 128 the sentence'target scale at the time and air domain' should probably sound: '.. in the time and angle domain'.

Response 4: Thank you for the comment. We fixed the writing error and we replaced 'target scale at the time and air domain' with 'characteristics target scale in the time and angle domain' in lines 158-159.

Point 5: Within the lines 170-175 the authors cite the foreign results of the signal analysis included in papers [18] & [19]. The description of presented Figure 9. is too short and moreover the cited figure has no labels at the axes.

Response 5: Thank you for the comment. We have added the description of the method (plate element method) used in the simulation of the submarine scattering model in lines 219-259. And we added a description of the intensity of the submarine target in Figure 9 in lines 259-264. Moreover, we modified Figure 9 to make the target intensity of each part of the submarine more obvious by matching the filtering gain and labeling the target intensity of each part in Figure 9 in lines 264-266.

Point 6: The description of the configuration target-torpedo stated in lines 185 - 186 is unclear and needs to be improved.

Response 6: Thank you for the comment. We have modified the description of the initial target-submarine position and motion in the simulation in lines 271-273.

Thank you for reading our responses and reviewing the revised manuscript!

Best regards.

Yours sincerely,

Zirui Wang.

Name: Zirui Wang.

E-mail: [email protected].

Reviewer 3 Report

The level of english has been improved on the manuscript. However the manuscript is not well organized. The comparison with the existing literature and the methods are not clearly presented. Novelty respect to the state of the art is not evidenced. The conclusions are weak and not supported by the results.

The manuscript must be rejected.

Author Response

Dear Prof:

Thank you for your very insightful comment. This letter records our responses to your comments on our manuscript entitled "A torpedo target recognition method based on the correlation between echo-broadening and apparent angle" (ID: applsci-2038914).

Point 1: The level of english has been improved on the manuscript. However the manuscript is not well organized. The comparison with the existing literature and the methods are not clearly presented.

Response 1: Thank you for your very insightful comment. We have added references to the existing underwater target scale recognition methods to illustrate the shortcomings of the current underwater target scale recognition by target intensity, echo-broadening, and apparent angle in lines 43-77, which leads to the proposed optimal target scale recognition method.

Point 2: Novelty respect to the state of the art is not evidenced.

Response 2: Thank you for your very valuable comment. We have added a comparison of the proposed target scale recognition method with the recognition results of current techniques and demonstrated that the echo-broadening and the apparent angle could not distinguish the scale difference between the real and virtual-scale targets in lines 322-331. And we have shown that the proposed method has better accuracy in target scale recognition under strong interference in lines 332-376.

Point 3: The conclusions are weak and not supported by the results.

Response 3: Thank you for your very insightful comment. Based on the simulation results, we have summarized the underwater target scale recognition method proposed in this paper, the scale recognition effect of the technique, and the scope of application of the method in lines 377-402. Moreover, we have presented the future outlook in lines 403-407. In addition, we have added a description of the signal processing of increase, time delay, and Doppler shift in acoustic decoy echo simulations using the highlight method in lines 190-218. We have added the description of the method (plate element method) used in the simulation of the submarine scattering model in lines 219-259. We have added an explanation of the propagation process of the acoustic signal in the ocean channel in lines 289-312, and we have added Figures 11 and 12 to depict the sound speed profile and the propagation process of the acoustic signal in the ocean channel, respectively, as applied in the simulation.

Thank you for reading our responses and reviewing the revised manuscript!

Best regards.

Yours sincerely,

Zirui Wang.

Name: Zirui Wang.

E-mail: [email protected].

Reviewer 4 Report

The paper gives some idea of how anti-submarine warfare works, how submarines are protected from it. Finally, the method is proposed for anti-submarine warfare to get through this protection. The idea is good, but the paper should have been written in a more academic way in order to be published in an academic journal. The more detailed description of the detected problems follows. 

1.              Many conclusions are supported with references to proceedings. It is true that the most recent research results are printed in proceedings first. However, the paper belongs to an area with a long history of study, so there some ‘stable’ results in this area. I feel that references to full-length journal papers, handbooks and tutorials are going to be more helpful to the readers since they contain extended and polished texts. Please, add references to journals and books wherever its possible.

2.              Please, give more detailed description of the submarine scattering model. Namely what is said in lines 24-28 is a too simplified summary of the long research, carried out in this area. What is said in lines 168-169 is too brief. Just giving the references is not enough.

3.              Please, give more detailed description of how you simulated sound propagation in the sea. What is said in lines 192-196 is too brief. Just giving the references is not enough.

4.              Please, give more detailed information: which signal processing algorithm do you assume to be exploited inside the simulated decoys.

5.              The authors are advised to think of a broader impact of their results. Maybe they have solved some mathematical problem that is more general that the described application. Maybe they have created a numerical code that could be used for simulations in underwater acoustics.

The submitted paper is too specialized on defense technology. Despite the journal is called ‘Applied science’, it might be not the best option for publishing this paper.

P.S. Sorry for my English

Author Response

Dear Prof:

Thank you for your very insightful comment. This letter records our responses to your comments on our manuscript entitled "A torpedo target recognition method based on the correlation between echo-broadening and apparent angle" (ID: applsci-2038914).

Point 1: The paper gives some idea of how anti-submarine warfare works, how submarines are protected from it. Finally, the method is proposed for anti-submarine warfare to get through this protection. The idea is good, but the paper should have been written in a more academic way in order to be published in an academic journal. The more detailed description of the detected problems follows:

Many conclusions are supported with references to proceedings. It is true that the most recent research results are printed in proceedings first. However, the paper belongs to an area with a long history of study, so there some ‘stable’ results in this area. I feel that references to full-length journal papers, handbooks and tutorials are going to be more helpful to the readers since they contain extended and polished texts. Please, add references to journals and books wherever its possible.

Response 1: Thank you for your very insightful comment. We have added references to full-length journal papers, handbooks and tutorials so that we can expand and embellish the text of this paper.

Point 2: Please, give more detailed description of the submarine scattering model. Namely what is said in lines 24-28 is a too simplified summary of the long research, carried out in this area. What is said in lines 168-169 is too brief. Just giving the references is not enough.

Response 2: Thank you for your valuable comment. We have added a summary of current underwater target scale recignition studies based on ‘stable’ research results in lines 26-40. We have added the description of the method (plate element method) used in the simulation of the submarine scattering model in lines 219-259. And we added a description of the intensity of the submarine target in Figure 9 in lines 259-264. Moreover, we modified Figure 9 to make the target intensity of each part of the submarine more obvious by matching the filtering gain and labeling the target intensity of each part in Figure 9 in lines 264-266.

Point 3: Please, give more detailed description of how you simulated sound propagation in the sea. What is said in lines 192-196 is too brief. Just giving the references is not enough.

Response 3: Thank you for your very insightful comment. We have added an explanation of the propagation process of the acoustic signal in the ocean channel in lines 289-312, and we have added Figures 11 and 12 to depict the sound speed profile and the propagation process of the acoustic signal in the ocean channel, respectively, as applied in the simulation.

Point 4: Please, give more detailed information: which signal processing algorithm do you assume to be exploited inside the simulated decoys.

Response 4: Thank you for your valuable comment. We have added a description of the signal processing of increase, time delay, and Doppler shift in acoustic decoy echo simulations using the highlight method in lines 190-218.

Point 5: The authors are advised to think of a broader impact of their results. Maybe they have solved some mathematical problem that is more general that the described application. Maybe they have created a numerical code that could be used for simulations in underwater acoustics.

The submitted paper is too specialized on defense technology. Despite the journal is called ‘Applied science’, it might be not the best option for publishing this paper.

Response 5: Thank you for your very insightful comment. We studied the accurate target scale recognition method by torpedoes in strong jamming. This method considers harsh and extreme situations and can also be applied to the recognition of other ships and reefs by active sonar of vessels and fish size recognition by fishing boats. We add a description of the more general application of the method in lines 21-23, 87-88 and 399-407.

Thank you for reading our responses and reviewing the revised manuscript!

Best regards.

Yours sincerely,

Zirui Wang.

Name: Zirui Wang.

E-mail: [email protected].

Reviewer 5 Report

Globally, the manuscript is very well written and organized. However, there are some English errors that must be corrected. I recommend the readers to do a deep global reading of the manuscript; please also refer to the attached commented PDF file, where some of these errors are highlighted.

A comparison between the results obtained by the method proposed here and the ones obtained by the currently used methods (state of the art) would be very interesting and informative. This would help understand what is the “real” improvement of the proposed method.

Lines 93-94: the meaning of “c” should also be include in this explanation.

Line 198: the number of the figure the authors are referring to should be included.

Author Response

Dear Prof:

Thank you for your very insightful comment. This letter records our responses to your comments on our manuscript entitled "A torpedo target recognition method based on the correlation between echo-broadening and apparent angle" (ID: applsci-2038914). And this letter also includes the manuscript with the track changes version.

Point 1: Globally, the manuscript is very well written and organized. However, there are some English errors that must be corrected. I recommend the readers to do a deep global reading of the manuscript; please also refer to the attached commented PDF file, where some of these errors are highlighted.

A comparison between the results obtained by the method proposed here and the ones obtained by the currently used methods (state of the art) would be very interesting and informative. This would help understand what is the “real” improvement of the proposed method.

Lines 93-94: the meaning of “c” should also be include in this explanation.

Response 1: Thank you for the comment. We have added the explanation of the parameter c. The modified parameters c in the article indicate the underwater sound speed in lines 124-125 and 185.

Point 2: Line 198: the number of the figure the authors are referring to should be included.

Response 2: Thank you for the comment. We have added the image numbers corresponding to each image in line 316 and corrected the other errors you pointed out in the pdf, Professor.

Thank you for reading our responses and reviewing the revised manuscript!

Best regards.

Yours sincerely,

Zirui Wang.

Name: Zirui Wang.

E-mail: [email protected].

Reviewer 6 Report

The authors presented a better work. However, authors may consider the following. 

1. The abstract requires modification to provide a comparison of the results with the benchmarking. 

2. Thorough proofreading is recommended

3. Figure 11 is not easily readable

4. A few of the references are missing some information, you may complete them critically.

5. Future work may be added in the Conclusion section. 

Author Response

Dear Prof:

Thank you for your very insightful comment. This letter records our responses to your comments on our manuscript entitled "A torpedo target recognition method based on the correlation between echo-broadening and apparent angle" (ID: applsci-2038914).

Point 1: The authors presented a better work. However, authors may consider the following:

The abstract requires modification to provide a comparison of the results with the benchmarking.

Response 1: Thank you for the comment. We have added the  comparison of the results with the benchmarking in the abstract in lines 17-23.

Point 2: Thorough proofreading is recommended..

Response 2: Thank you for the comment. We have proofread this article and fixed the errors in the article.

Point 3: Figure 11 is not easily readable.

Response 3: Thank you for the comment. We have added Figures 11 and 12 to depict the sound speed profile and the propagation process of the acoustic signal in the ocean channel, respectively. And we have replaced the original numbering of Figure 11 with Figure 13. We have adjusted the image scale and text layout of Figure 13 in lines 314-315 so that it is not deformed and can be read more clearly.

Point 4: A few of the references are missing some information, you may complete them critically.

Response 4: Thank you for the comment. We have completed the reference information through endnote software.

Point 5: Future work may be added in the Conclusion section.

Response 5: Thank you for the comment. We have added plans for future work that will help us conduct more in-depth research in lines 403-407.

Thank you for reading our responses and reviewing the revised manuscript!

Best regards.

Yours sincerely,

Zirui Wang.

Name: Zirui Wang.

E-mail: [email protected].

Round 2

Reviewer 3 Report

The article seems to be better structured and cared for from a scientific point of view. I think it can be published in this form.

Author Response

Dear Prof:

Thank you for your encouraging comment. This letter records our responses to your comments on our manuscript entitled "A torpedo target recognition method based on the correlation between echo-broadening and apparent angle" (ID: applsci-2038914).

Point 1: The article seems to be better structured and cared for from a scientific point of view. I think it can be published in this form.

Response 1: Thank you for your encouraging comment! We have modified the equation 13 boundary condition in lines 233-236, where φs = φi indicates that the incident potential function in the rigid interface condition is equal in strength to the scattering potential function. We have reinterpreted Eq. 14 in lines 238-239, where V(θ) is 1 for each angle in the rigid interface condition, so the scattering potential function expression does not contain V(θ). Moreover, we added the interpretation of the potential scattering function for non-rigid interface conditions in lines 239-252, which includes V(θ) in this expression.

Thank you for reading our responses and reviewing the revised manuscript!

Best regards.

Yours sincerely,

Zirui Wang.

Name: Zirui Wang.

E-mail: [email protected].

Reviewer 4 Report

The reviewer appreciates the fact that the author has addressed all arisen issues. The information that the author has inserted is very helpful for the reader and it revealed that the study was conducted very thoroughly. One more comment.

Eq. (13) is titled as rigid boundary conditions. To be more precise, only the line with d/dn(…)=0 is the rigid boundary conditions. The line with phis=phii is result of the Kirgchoff approximation that was applied here.

Later it is said that these boundary conditions lead to (14), and (14) contains V(theta). In fact the conditions (13) corresponds to V(theta)=1 only. The model with variable V(theta) is convinient, but the explanation is not exact in that passage.

Author Response

Dear Prof:

Thank you for your very insightful comment. This letter records our responses to your comments on our manuscript entitled "A torpedo target recognition method based on the correlation between echo-broadening and apparent angle" (ID: applsci-2038914).

Point 1: The reviewer appreciates the fact that the author has addressed all arisen issues. The information that the author has inserted is very helpful for the reader and it revealed that the study was conducted very thoroughly. One more comment.

Eq. (13) is titled as rigid boundary conditions. To be more precise, only the line with d/dn(…)=0 is the rigid boundary conditions. The line with phis=phii is result of the Kirgchoff approximation that was applied here.

Later it is said that these boundary conditions lead to (14), and (14) contains V(theta). In fact the conditions (13) corresponds to V(theta)=1 only. The model with variable V(theta) is convinient, but the explanation is not exact in that passage.

Response 1: Thank you for your very insightful comment.  We have modified the equation 13 boundary condition in lines 233-236, where φs = φi indicates that the incident potential function in the rigid interface condition is equal in strength to the scattering potential function. We have reinterpreted Eq. 14 in lines 238-239, where V(θ) is 1 for each angle in the rigid interface condition, so the scattering potential function expression does not contain V(θ). Moreover, we added the interpretation of the potential scattering function for non-rigid interface conditions in lines 239-252, which includes V(θ) in this expression.

Thank you for reading our responses and reviewing the revised manuscript!

Best regards.

Yours sincerely,

Zirui Wang.

Name: Zirui Wang.

E-mail: [email protected].
